# US adults with diabetes mellitus: Variability in oral healthcare utilization

Lorena Baccaglini[1], Adams Kusi Appiah[2], Mahua Ray[2], Fang Yu[2]*

1 Department of Epidemiology, College of Public Health, University of Nebraska Medical Center, Omaha, NE, United States of America, 2 Department of Biostatistics, College of Public Health, University of Nebraska Medical Center, Omaha, NE, United States of America

* fangyu@unmc.edu

## Abstract

### Background

Diabetic patients are advised to have at least one dental examination per year. It is unclear to what extent different subgroups of US diabetic adults closely follow this recommendation. Thus, we assessed dental care utilization and related factors in a representative sample of US diabetic adults from rural and urban counties.

### Methods

Cross-sectional data were from the 2018 Behavioral Risk Factor Surveillance System (BRFSS). Survey logistic regression was used to account for the complex sampling design.

### Results

Among 40,585 eligible participants, 24,887 (60% of the population) had at least one dental visit for any reason within the past year. The lowest compliance was observed among edentulous participants (27%, adjusted OR = 0.26, 95% CI = 0.22–0.31 vs. fully dentate). Dental compliance was also negatively associated with having a lower income or education, ever being a smoker, or having barriers to access to care. Rural residents had lower dental compliance compared to urban residents, particularly those without healthcare coverage.

### Conclusions

Dental compliance among US adult diabetic individuals was low, particularly among rural residents, and as compared to other recommended diabetic care practices. Future public health interventions may target rural individuals without healthcare coverage, smokers and edentulous individuals. There is a need to integrate dental and medical care to facilitate cross-talks among different health professionals, so that educational preventive messages are reinforced at every healthcare visit.

**Data Availability Statement:** The data are available from the BRFSS data repository (https://www.cdc.gov/brfss/annual_data/annual_2018.html).

**Funding:** This work was partially supported by the National Institutes of Health, National Institute of

General Medical Sciences (# 5U54GM115458).
There was no additional external funding received
for this study. The content is solely the
responsibility of the authors and does not
necessarily represent the official views of the
National Institutes of Health.

**Competing interests:** The authors have declared
that no competing interests exist.

## Introduction

The number of diabetes cases worldwide is expected to grow to 4.4% by 2030 from just 2.8% in 2000 [1]. More than 90% of diabetic patients have oral manifestations, ranging from dental caries to xerostomia, periodontal disease, sensory disorders, taste problems, salivary gland dysfunction, and oral infections [2].

Diabetes and oral disease share a bidirectional relationship. Acute oral complications are commonly seen in poorly controlled diabetic patients [3–6]. Diabetic patients are more likely to develop periodontal disease compared to non-diabetic patients, and periodontitis progresses more rapidly in poorly controlled diabetic individuals [7,8].

Severe periodontal disease may be a warning sign of undiagnosed diabetes [9]. Periodontal treatment can reduce glycemia among diabetic patients [10].

The Centers for Disease Control and Prevention (CDC), and the American Diabetes Association (ADA) recommend that diabetic patients have a dental examination at least once every year [11,12]. According to Healthy People 2020, the goal is that 61.2% or more of people with diabetes have an annual dental examination; however, in 2008 the age-adjusted percentage of US dentate diabetic patients who had been to the dentist within the past year was 55.6% [13].

Regular dental visits are useful not only to screen, diagnose and treat oral diseases, but they can also serve as an additional opportunity to provide counseling to diabetic patients, such as emphasizing the need for regular blood glucose checks and maintaining appropriate glycemic control [14]. Dental visits may also identify diabetic or pre-diabetic patients unaware of their risk or condition [15].

Prior studies have shown that the frequency of dental visits among diabetic patients is directly related to patients' income level, education, and access to affordable insurance [16–18]. However, African Americans or Hispanics with diabetes, or diabetic individuals with fewer natural teeth are less likely to have had a dental visit within the past year [17,18]. Additionally, prior studies of the general US adult population have reported that rural residents have lower odds of having received preventive dental procedures or having had a dental visit in the past year compared to metropolitan residents [19,20].

Although diabetic patients are advised to have a dental examination at least annually, it is unclear to what extent different subgroups of US diabetic adults from rural and urban counties closely follow this recommendation. Thus, we assessed dental care utilization and identified specific factors associated with annual dental visits in a representative sample of US diabetic adults overall, as well as among rural and urban residents.

## Materials and methods

We analyzed the 2018 Behavioral Risk Factor Surveillance System (BRFSS; https://www.cdc.gov/brfss/annual_data/annual_2018.html), a phone-based national survey of the civilian, noninstitutionalized adult population aged 18 or older conducted by the CDC using a complex multistage probability sample [21]. The analyses included participants with diabetes mellitus and complete data on the variables of interest. Individuals with gestational diabetes, pre-diabetes, or borderline diabetes were excluded.

The outcome variable was dental care compliance, defined as having visited a dentist, dental hygienist or dental clinic for any reason within the past year. Independent variables included self-reported information on demographic, socio-economic, health and access-to-care factors, and living in a rural or urban county. Demographic factors were age, sex and race/ethnicity. Socioeconomic factors included marital status, annual household income, education, employment status, smoking status, exercise in the past 30 days, and average number of alcoholic drinks per day in the past 30 days. Health factors included self-reported current general health,

poor mental health (stress, depression and problems with emotions during the past 30 days), poor physical health (physical illness and injury during the past 30 days) and Body Mass Index (BMI). Access-to-care factors were healthcare coverage (including health insurance, prepaid plans such as HMOs, or government plans such as Medicare, or Indian Health Service), having a personal doctor, and perceiving cost as a barrier to medical care.

To account for the complex, stratified, multistage sampling design we used the SAS (Version 9.4, SAS Institute, Inc, Cary, NC) procedures SURVEYFREQ, and SURVEYLOGISTIC. These procedures incorporate survey design variables (weights, strata, and primary sampling units) in the analyses to obtain nationally representative estimates. Counts and percentages were used for descriptive statistics. Univariable and multivariable logistics regression models were used to calculate crude or adjusted odds ratio (OR or AOR) with 95% CI for the association between dental compliance and the independent variables. The final model was identified via backward variable selection using p>0.1 for removal. We also conducted separate sub-analyses by urban and rural residence using predictors identified in the full model.

## Results

The final number of eligible participants was 40,585 (Fig 1). Sixty-one percent of participants (corresponding to 19,281,982 or 60% of the US population) had at least one dental visit within the past year.

A lower percentage of Hispanics (51%) and non-Hispanic Blacks (55%) had an annual dental visit compared to non-Hispanic Whites (62%; Table 1). However, after adjusting for other variables, compliance was similar across different race/ethnicity and age groups (Table 2). Females had 18% higher adjusted odds of having had at least one dental visit within the past year compared to males.

Participants with good or better general health compared to those with fair or poor health were also more compliant (AOR = 1.23; 95% CI = 1.10–1.36). Fully edentulous participants had the lowest (27%) compliance compared to dentate participants (55% or higher; AOR = 0.26, 95% CI = 0.22–0.31 comparing to fully dentate). Individuals who had ever smoked (i.e., current or former smokers) had lower crude and adjusted odds of compliance compared to never smokers (current: AOR = 0.69, 95% CI = 0.59–0.81; former: AOR = 0.92, 95% CI = 0.83–1.03).

Participants with the highest income or education had the highest crude and adjusted odds (AOR>1.9) of reporting a dental visit within the past year compared to their counterparts (Table 2). A lower proportion of participants who reported no personal doctor (43% vs. 61%), no healthcare coverage (36% vs. 61%) or medical cost as a barrier to seeing a doctor (43% vs. 62%) were compliant compared to those with better access to medical care. Lack of healthcare coverage was a particularly strong indicator of low compliance with dental care visits, especially among rural (24% compliance) vs. urban residents (37%), even after adjusting for other factors (Fig 2).

Overall, urban participants had higher adjusted odds of compliance compared to their Rural counterparts (AOR = 1.17; 95% CI = 1.01–1.35), with some notable differences (Fig 2 and S1 Table). Specifically, among urban participants, obesity was negatively associated with dental care utilization (59% of obese individuals had a dental visit within the past year vs. 62% of other urban residents), whereas a higher percentage (53%) of obese rural residents had a dental visit compared to other rural subgroups (43% of normal/underweight and 49% of overweight participants).

Individuals living as a couple (married or living together as married) in either urban or rural areas were more compliant than other groups (Fig 2). The least compliant groups were

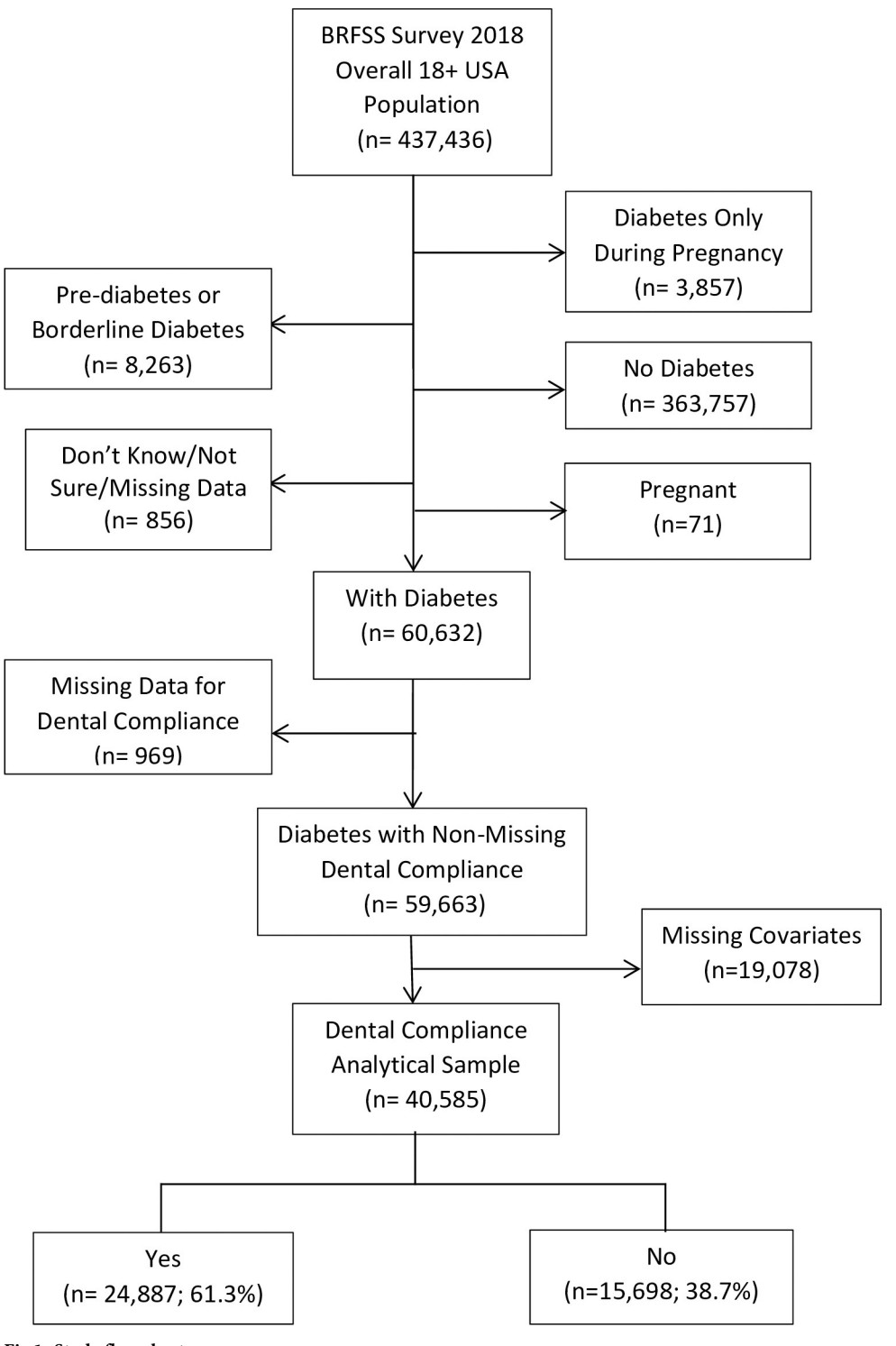

**Fig 1. Study flow chart.**

widows in urban areas (52% compliance) and singles (divorced, separated, or never married) in rural areas (38%).

Urban residents with higher income also had higher adjusted odds of compliance, whereas this association was weaker and not significant in the rural stratum. Specifically, urban

**Table 1. Dental visit compliance by participants' characteristics.**

| Characteristics | Dental visit compliance (n = 40,585; N = 19,281,982) | | | |
| --- | --- | --- | --- | --- |
| | Yes (n = 24,887; N = 11,506,723) | | No (n = 15,698; N = 7,775,260) | |
| | Sample frequency | Population estimate | Sample frequency | Population estimate |
| | n | N (thousand; %)* | n | N (thousand; %)* |
| Age (years) | | | | |
| 18–44 | 1,749 | 1,527.2 (60.0) | 1,208 | 1,018.9 (40.0) |
| 45–64 | 9,752 | 5,148.3 (58.7) | 6,527 | 3,629.5 (41.3) |
| 65 or older | 13,386 | 4,831.2 (60.7) | 7,963 | 3,126.8 (39.3) |
| Sex | | | | |
| Male | 12,738 | 5,224.5 (60.4) | 7,837 | 4,122.0 (39.6) |
| Female | 12,149 | 11,506.7 (58.8) | 7,861 | 3,653.2 (41.2) |
| Race/Ethnicity | | | | |
| Non-Hispanic (NH) White | 18,843 | 7,510.9 (62.0) | 11,122 | 4,603.8 (38.0) |
| NH Black | 2,674 | 1,558.4 (54.6) | 2,186 | 1,294.2 (45.4) |
| NH Other/Multiracial | 1,906 | 964.7 (66.5) | 1,252 | 485.4 (33.5) |
| Hispanic | 1,464 | 1,472.7 (51.4) | 1,138 | 1,391.8 (48.6) |
| Marital status | | | | |
| Couple | 14,374 | 7,229.3 (65.1) | 6,848 | 3,869.8 (34.9) |
| Divorced/Separated/Never married | 6,689 | 3,034.1 (52.4) | 5,750 | 2,751.0 (47.6) |
| Widowed | 3,824 | 1,243.3 (51.9) | 3,100 | 1,154.5 (48.1) |
| Education | | | | |
| < High school | 1,484 | 1,382.2 (41.1) | 2,394 | 1,978.3 (58.9) |
| High school graduate | 6,473 | 2,967.8 (53.2) | 6,015 | 2,608.5 (46.8) |
| Some college/Technical | 7,585 | 3,990.0 (63.6) | 4,557 | 2,283.4 (34.6) |
| College graduate | 9,345 | 3,166.7 (77.8) | 2,732 | 905.0 (22.2) |
| Income ($) | | | | |
| <15,000 | 2,287 | 1,203.9 (41.4) | 3,474 | 1,706.1 (58.6) |
| 15,000–34,999 | 6,838 | 3,080.3 (48.4) | 6,948 | 3,277.8 (51.6) |
| 35,000–49,999 | 3,843 | 1,461.5 (60.0) | 2,025 | 973.3 (40.0) |
| ≥50,000 | 11,919 | 5,761.1 (76.0) | 3,251 | 1,818.1 (24.0) |
| Employment | | | | |
| Employed | 8,809 | 4,689.4 (65.7) | 4,065 | 2,452.1 (34.3) |
| Homemaker/Student/Retired | 12,684 | 4,974.3 (62.0) | 7,210 | 3,045.6 (38.0) |
| Not employed/Unable to work | 3,394 | 1,843.0 (44.7) | 4,423 | 2,277.6 (55.3) |
| Healthcare coverage[†] | | | | |
| Yes | 24,144 | 11,029.5 (61.4) | 14,597 | 6,921.3 (38.6) |
| No | 743 | 477.2 (35.8) | 1,101 | 854.0 (64.2) |
| Personal doctor | | | | |
| Yes | 23,841 | 10,949.9 (60.9) | 14,517 | 7,023.3 (39.1) |
| No | 1,046 | 556.8 (42.5) | 1,181 | 752.0 (57.5) |
| Medical cost[$] | | | | |
| Yes | 1,899 | 1,114.5 (42.9) | 2,582 | 1,483.6 (57.1) |
| No | 22,988 | 10,392.3 (62.3) | 13,116 | 6,291.7 (37.7) |
| Self-reported general health | | | | |
| Good/Better | 16,496 | 7,354.0 (66.9) | 7,611 | 3,631.7 (33.1) |
| Fair/Poor | 8,391 | 4,152.8 (50.1) | 8,087 | 4,143.5 (49.9) |
| Number of permanent teeth removed | | | | |
| None | 8,811 | 4,294.1 (68.8) | 3,438 | 1,946.1 (31.2) |

*(Continued)*

**Table 1.** (Continued)

| Characteristics | Dental visit compliance (n = 40,585; N = 19,281,982) | | | |
| --- | --- | --- | --- | --- |
| | Yes (n = 24,887; N = 11,506,723) | | No (n = 15,698; N = 7,775,260) | |
| | Sample frequency | Population estimate | Sample frequency | Population estimate |
| | n | N (thousand; %)* | n | N (thousand; %)* |
| 1–5 | 9,703 | 4,467.2 (64.4) | 4,225 | 2,466.5 (35.6) |
| 6+ | 5,022 | 2,145.8 (54.9) | 3,857 | 1,761.9 (45.1) |
| All | 1,351 | 599.6 (27.2) | 4,178 | 1,600.8 (72.8) |
| Smoking status[II] | | | | |
| Current | 2,535 | 1,312.1 (45.2) | 3,194 | 1,589.8 (54.8) |
| Former | 8,964 | 3,931.3 (59.3) | 5,834 | 2,701.0 (40.7) |
| Never | 13,388 | 6,263.3 (64.3) | 6,670 | 3,484.5 (35.7) |
| Alcohol intake in past 30 days | | | | |
| None | 14,919 | 6,813.8 (55.9) | 11,253 | 5,366.2 (44.1) |
| 1 drink/day | 5,351 | 2,269.2 (69.6) | 2,108 | 991.6 (30.4) |
| 2+ drinks/day | 4,617 | 2,423.8 (63.1) | 2,337 | 1,417.5 (36.9) |
| Poor mental health in past 30 days | | | | |
| Yes | 7,843 | 3,728.0 (54.0) | 6,375 | 3,172.8 (46.0) |
| No | 17,044 | 7,778.7 (62.8) | 9,323 | 4,602.4 (37.2) |
| Poor physical health in past 30 days | | | | |
| Yes | 12,157 | 5,645.8 (54.9) | 9,356 | 4,642.6 (45.1) |
| No | 12,730 | 5,860.9 (65.2) | 6,342 | 3,132.6 (34.8) |
| Exercise in past 30 days[¶] | | | | |
| Yes | 16,891 | 7,929.5 (64.7) | 8,482 | 4,329.7 (35.3) |
| No | 7,996 | 3,577.2 (50.9) | 7,216 | 3,445.6 (49.1) |
| Body Mass Index (BMI) [#] | | | | |
| Underweight/Normal weight | 3,368 | 1,630.5 (60.9) | 2,104 | 1,045.1 (39.1) |
| Overweight | 8,012 | 3,639.4 (61.6) | 4,515 | 2,265.0 (38.4) |
| Obese | 13,507 | 6,236.9 (58.3) | 9,079 | 4,465.1 (41.7) |
| Residence | | | | |
| Urban | 21,017 | 10,683.3 (60.5) | 12,577 | 6,974.8 (39.5) |
| Rural | 3,870 | 823.4 (50.7) | 3,121 | 800.4 (49.3) |

*Population estimates (N and %) are adjusted for the complex sampling design.

[†]Healthcare coverage: Has any kind of health care coverage, including health insurance, prepaid plans such as health maintenance organization (HMO), government plans such as Medicare, or Indian Health Service.

Personal doctor: Has a personal doctor or health care provider.

[§]Medical cost: Could not see a doctor because of cost.

[II]Smoking status: Current: Smokes every day or some days, Former: Smoked before, Never: Never smoked.

[¶]Exercise: Participated in any physical activities or exercises such as running, calisthenics, golf, gardening, or walking for exercise in the past 30 days.

[#]BMI; Underweight/Normal weight: BMI<25, Overweight: 25≤BMI <30, Obese: BMI≥30.

participants in the highest income bracket had 2.27 times (95% CI = 1.83–2.82) higher adjusted odds of compliance vs. those in the lowest income bracket.

## Discussion

In 2018, 60% of US adult diabetic participants reported at least one dental visit for any reason within the past year. Although our estimate is limited to adults only, the percentage is very close to the Healthy People 2020 target (61.2%) for all diabetic individuals aged 2 years or

**Table 2. Crude and adjusted odds ratios (OR) for dental visit compliance by participants' characteristics (n = 40,585; N = 19,281,982).**

| Characteristics | Univariable | Full model* | Final model* |
|---|---|---|---|
| | Crude OR (95% CI) | Adjusted OR (95% CI) | Adjusted OR (95% CI) |
| Age (years) | | | |
| 18–44 vs 65 or older | 0.97 (0.83, 1.13) | 1.10 (0.89, 1.35) | – |
| 45–64 vs 65 or older | 0.92 (0.84, 1.01) | 1.02 (0.89, 1.18) | |
| Sex | | | |
| Female vs Male | 0.94 (0.86, 1.03) | 1.19 (1.07, 1.33) | 1.18 (1.06, 1.31) |
| Race/Ethnicity | | | |
| Non-Hispanic (NH) Black vs NH White | 0.74 (0.65, 0.83) | 0.93 (0.81, 1.06) | – |
| Hispanic vs NH White | 0.65 (0.55, 0.76) | 0.90 (0.74, 1.10) | |
| NH Other/Multiracial vs NH White | 1.22 (0.98, 1.51) | 1.07 (0.83, 1.36) | |
| Marital Status | | | |
| Divorced/Separated/Never married vs Couple | 0.59 (0.53, 0.66) | 0.93 (0.81, 1.05) | – |
| Widowed vs Couple | 0.58 (0.51, 0.66) | 0.87 (0.75, 1.01) | |
| Education | | | |
| High school (HS) graduate vs <HS | 1.63 (1.40, 1.90) | 1.11 (0.93, 1.31) | 1.13 (0.96, 1.34) |
| Some college/Technical vs <HS | 2.50 (2.14, 2.92) | 1.33 (1.11, 1.59) | 1.36 (1.14, 1.61) |
| College graduate vs <HS | 5.01 (4.25, 5.90) | 1.85 (1.53, 2.24) | 1.91 (1.58, 2.31) |
| Income ($) | | | |
| 15,000–34,999 vs <15,000 | 1.33 (1.15, 1.54) | 1.07 (0.92, 1.25) | 1.10 (0.94, 1.28) |
| 35,000–49,999 vs <15,000 | 2.13 (1.81, 2.51) | 1.29 (1.06, 1.56) | 1.35 (1.12, 1.64) |
| ≥50,000 vs <15,000 | 4.49 (3.86, 5.22) | 2.17 (1.77, 2.65) | 2.33 (1.93, 2.82) |
| Employment | | | |
| Homemaker/Student/Retired vs Employed | 0.85 (0.77, 0.95) | 1.19 (1.04, 1.38) | 1.18 (1.05, 1.33) |
| Not employed/Unable to work vs Employed | 0.42 (0.37, 0.48) | 0.99 (0.84, 1.15) | 0.97 (0.83, 1.14) |
| Healthcare coverage[†] | | | |
| Yes vs No | 2.85 (2.34, 3.48) | 1.69 (1.36, 2.12) | 1.70 (1.36, 2.12) |
| Personal doctor | | | |
| Yes vs No | 2.11 (1.74, 2.55) | 1.42 (1.16, 1.75) | 1.42 (1.16, 1.74) |
| Medical cost[§] | | | |
| Yes vs No | 0.45 (0.40, 0.52) | 0.69 (0.59, 0.81) | 0.69 (0.59, 0.80) |
| Self-reported general health | | | |
| Good/Better vs Fair/Poor | 2.02 (1.84, 2.22) | 1.19 (1.07, 1.33) | 1.23 (1.10, 1.36) |
| Number of permanent teeth removed | | | |
| 1–5 vs None | 0.82 (0.73, 0.93) | 1.03 (0.91, 1.17) | 1.02 (0.90, 1.16) |
| 6+ vs None | 0.55 (0.49, 0.63) | 0.84 (0.73, 0.96) | 0.82 (0.72, 0.94) |
| All vs None | 0.17 (0.15, 0.20) | 0.27 (0.23, 0.32) | 0.26 (0.22, 0.31) |
| Smoking status[‖] | | | |
| Current vs Never | 0.46 (0.40, 0.53) | 0.69 (0.59, 0.82) | 0.69 (0.59, 0.81) |
| Former vs Never | 0.81 (0.73, 0.89) | 0.92 (0.83, 1.03) | 0.92 (0.83, 1.03) |
| Alcohol intake in past 30 days | | | |
| 1 drink/day vs None | 1.80 (1.60, 2.04) | 1.13 (0.99, 1.30) | – |
| 2+ drinks/day vs None | 1.35 (1.18, 1.53) | 0.96 (0.84, 1.11) | |
| Poor mental health in past 30 days | | | |
| Yes vs No | 0.70 (0.63, 0.76) | 0.93 (0.83, 1.04) | – |
| Poor physical health in past 30 days | | | |
| Yes vs No | 0.65 (0.59, 0.71) | 0.97 (0.87, 1.08) | – |
| Exercise in past 30 days[¶] | | | |

*(Continued)*

**Table 2.** (Continued)

| Characteristics | Univariable | Full model* | Final model* |
|---|---|---|---|
| | Crude OR (95% CI) | Adjusted OR (95% CI) | Adjusted OR (95% CI) |
| Yes vs No | 1.76 (1.61, 1.94) | 1.27 (1.14, 1.42) | 1.28 (1.15, 1.42) |
| Body Mass Index (BMI) # | | | |
| Underweight/Normal weight vs Obese | 1.12 (0.97, 1.28) | 1.16 (0.99, 1.35) | 1.17 (1.00, 1.36) |
| Overweight vs Obese | 1.15 (1.04, 1.28) | 1.14 (1.02, 1.27) | 1.14 (1.02, 1.27) |
| Residence | | | |
| Urban vs Rural | 1.49 (1.30, 1.70) | 1.20 (1.04, 1.37) | 1.17 (1.01, 1.35) |

*OR and 95% CI are adjusted for all the variables listed in the column, and the complex sampling design.

†Healthcare coverage: Has any kind of health care coverage, including health insurance, prepaid plans such as health maintenance organization (HMO), or government plans such as Medicare, or Indian Health Service.

Personal doctor: Has a personal doctor or health care provider.

§Medical cost: Could not see a doctor because of cost.

‖Smoking status: Current: Smokes every day or some days, Former: Smoked before, Never: Never smoked.

¶Exercise: Participated in any physical activities or exercises such as running, calisthenics, golf, gardening, or walking for exercise in the past 30 days.

#BMI; Underweight/Normal weight: BMI<25, Overweight: 25≤BMI <30, Obese: BMI≥30.

older [13]. However, our figure is lower than that previously reported for the adult US population in 2011–2014 (66%) and 2014 (68%) [22,23]. This finding is not unexpected, as prior studies have also reported lower compliance among diabetic individuals compared to the general population [24].

Compliance with annual dental visits was higher among diabetic females after adjusting for other factors. This finding is consistent with prior reports of higher female compliance in the general US population [23]. Diabetic individuals of different age, racial and ethnic groups

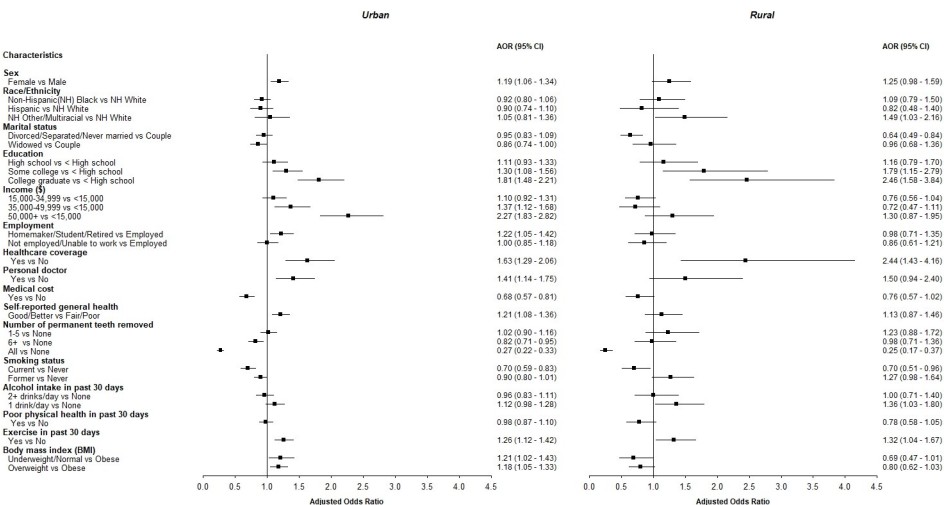

**Fig 2. Dental visit compliance by residential status.** Odds ratios (OR) and 95% CI for dental visit compliance by urban and rural residence status adjusted for sex, age, BMI, race/ethnicity, marital status, education, employment, smoking status, self-reported general health, healthcare coverage, having a personal doctor, perceiving medical cost as a barrier, exercise in the past 30 days, average alcohol intake in the past 30 days, poor mental or physical health in the past 30 days, annual household income, and number of permanent teeth removed (n = 40,585; N = 19,281,982). Only variables with a significant association for either residential status are shown. Full details are available in the supplemental table.

reported similar dental compliance after adjusting for other variables. It appears that differences among these groups are less pronounced among diabetics compared to the general population [23]. These findings differ from those of prior studies in which compliance was 9–22% lower among Hispanic vs. non-Hispanic adults with diabetes, and it was also lower among non-Hispanic Black versus non-Hispanic White diabetic individuals [17,24,25].

Overall, diabetic individuals with better general health and urban residents had higher adjusted odds of compliance (Table 2). Higher dental visits compliance among urban residents has also been reported for the general US population [19].

Edentulous participants were less compliant compared to dentate participants. A possible reason is that edentulous individuals may be less aware of the need to continue oral check-ups ever after all the teeth have been removed. Lack of general awareness among diabetic patients regarding increased susceptibility to oral diseases and their impact on overall health conditions was noted in a recent study [26].

Diabetic individuals with a lifetime history of smoking had lower crude and adjusted odds of dental compliance compared to never smokers, following a similar pattern as that previously reported for the general population [23]. This population subgroup may require targeted preventive efforts because smoking is also a risk factor for poorer dental outcomes [27].

Reduced access to medical care (e.g., having no personal doctor, or perceiving medical cost as a barrier) was associated with lower compliance among both rural and urban residents. Prior studies have found that income is associated with higher dental visit compliance among diabetic dentate adults [18]. We observed a similar trend among urban residents. However, income was not linked to compliance in rural populations after adjusting for other factors. Having lower education or no health insurance coverage were associated with lower odds of dental compliance, as expected based on prior studies of both the diabetic and the general population [18,23]. Notably, the links between dental visits compliance and education or health insurance were more pronounced among rural diabetic residents. These findings suggest the need for more robust prevention efforts in rural areas, such as enhanced dental public health education.

Urban widows and single rural participants were less compliant than individuals living as a couple (married or living as married; Fig 2). A stronger social structure may contribute to improved knowledge, awareness, and encouragement toward better dental compliance. Thus, a possible explanation for this finding is that single individuals may be less socially active in rural vs. urban environments, whereas widows may have less family support in urban vs. rural environments.

The relationship between obesity and compliance differed by residence status, with urban obese residents being less compliant and rural obese residents more compliant than non-obese individuals after adjusting for other factors (Fig 2). Individuals with lower BMI tend to have lower dental disease burden [28]. Compliance with dental visits was lowest among non-obese rural residents, which suggests that this subgroup may not adequately seek preventive dental care. More in-depth studies are needed to investigate the opposite direction of association seen in rural vs. urban participants.

A limitation of this study was the use of self-reports collected at one time point. There was also no information indicating whether dental visits were for preventive care, emergencies or planned dental treatment. Thus, we expect the percentage of annual preventive dental visits to be even lower than our reported estimate. Some survey questions were more closely related to medical care (such as health insurance coverage or healthcare cost and barriers), although still applicable to dental care. Strengths of this study were the use of standardized data collection procedures and the large, nationally representative sample of US adults, although certain subgroups were not represented (e.g., institutionalized individuals, military, and pregnant women).

## Conclusions

Avoidance of oral infections through optimal dental care and routine dental visits is paramount to reaching and maintaining good glycemic control and minimizing diabetic complications. However, a high percentage (40%) of US adult diabetic individuals reported no dental visits within the past year. This estimate is substantially higher than the percentage of diabetic individuals not following other yearly preventive care practices, such as annual eye exams (30%) or annual cholesterol checks (7%) [29].

Despite the bidirectional relationship between oral health and diabetes, collaboration between the medical and dental professions is limited. There is a need to integrate dental and medical care to facilitate cross-talks among different health professionals, so that educational preventive messages are reinforced at every healthcare visit [30]. Prior studies have shown that compliance with preventive dental visits by diabetic patients is reinforced through advisement of health care professionals [16]. The integration of dental and medical messages may be facilitated by the explicit inclusion of preventive dental care practices in the ADA's diabetes care practice guidelines.

Overall, the study findings suggest additional potential areas for improvement, such as improved public health educational messages that encourage preventive dental visits among individuals who smoke, are edentulous or have no healthcare coverage. Public health interventions and messages may need to be customized to reach different rural and urban subgroups that underutilize dental services.

## Supporting information

**S1 Table. Dental visit compliance by urban and rural participants' characteristics.**
(DOCX)

## Acknowledgments

The authors would like to thank Zaeema Naveed for assisting with manuscript formatting.

## Author Contributions

**Conceptualization:** Lorena Baccaglini, Mahua Ray, Fang Yu.

**Formal analysis:** Adams Kusi Appiah, Mahua Ray, Fang Yu.

**Methodology:** Lorena Baccaglini, Mahua Ray, Fang Yu.

**Writing – original draft:** Lorena Baccaglini, Fang Yu.

**Writing – review & editing:** Lorena Baccaglini, Adams Kusi Appiah, Mahua Ray, Fang Yu.

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
