## [Decision Letter · Decision Letter 0]

9 Apr 2021

PONE-D-21-04886

US adults with diabetes mellitus: Variability in oral healthcare utilization

PLOS ONE

Dear Dr. Yu,

Thank you for submitting your manuscript to PLOS ONE. After careful consideration, we feel that it has merit but does not fully meet PLOS ONE’s publication criteria as it currently stands. Therefore, we invite you to submit a revised version of the manuscript that addresses the points raised during the review process.

We look forward to receiving your revised manuscript.

Kind regards,

Denis Bourgeois

Academic Editor

PLOS ONE

Journal Requirements:

Thank you for stating in your Funding Statement:

This work was partially supported by the National Institutes of Health, National Institute of General Medical Sciences (# 5U54GM115458).

Reviewers' comments:

Reviewer's Responses to Questions

**Comments to the Author**

1. Is the manuscript technically sound, and do the data support the conclusions?

Reviewer #1: Partly

Reviewer #2: Yes

2. Has the statistical analysis been performed appropriately and rigorously? 

Reviewer #1: Yes

Reviewer #2: Yes

3. Have the authors made all data underlying the findings in their manuscript fully available?

Reviewer #1: Yes

Reviewer #2: Yes

4. Is the manuscript presented in an intelligible fashion and written in standard English?

Reviewer #1: Yes

Reviewer #2: Yes

5. Review Comments to the Author

Reviewer #1: Review Question 1: The author uses the term annual dental "visit" when it should state annual dental "examination" . The Healthy People 2020 Survey uses "annual dental examination" as language for both target and target setting method. The term "visit " is unclear as it could include planned or emergency dental treatment. Furthermore, the CDC Diabetes Care Schedule's states dental exam and cleaning at least once per year, and the ADA Standards of Medical Care in Diabetes (2020) state annual dental visit, but in the context of preventive care services (pp 47-48).

The author notes this limitation in the discussion section of the study, "There was also no information indicating whether dental visits were for preventive care, emergencies, or planned dental treatment." The author confuses the reader by using the word "visit" and not "examination" throughout the paper, making the discussion unclear and weakens the study's conclusions.

Revisiting to the CDC and ADA literature cited in the study and using the correct language throughout the paper may make the submission acceptable for publication. Recommendation: Major Revision

Reviewer #2: The methodology and statistical analysis are rigorous.

Authors must put the website address on which the BRFSS database is accessible.

The authors need to elaborate on the procedure of matching the sample to the US population.

The BMI must be defined.

6. PLOS authors have the option to publish the peer review history of their article (what does this mean?). If published, this will include your full peer review and any attached files.

Reviewer #1: **Yes: **Dr Martin R Gillis, DDS

Reviewer #2: No

---

## [Author Response · Author response to Decision Letter 0]

13 Apr 2021

Thank you for the opportunity to submit a revised draft of our manuscript “PONE-D-21-04886; US adults with diabetes mellitus: Variability in oral healthcare utilization”. We appreciate the reviewers’ insightful comments and suggestions on our paper. We have addressed all the comments and made the suggested changes (tracked) in the manuscript. We also revised our funding statement to indicate that “This work was partially supported by the National Institutes of Health, National Institute of General Medical Sciences (# 5U54GM115458). There was no additional external funding received for this study.” We believe that the paper is improved as a result of the revisions. We hope that our edits to the paper and the responses below satisfactorily address all the critiques.

Reviewer #1:

The author uses the term annual dental "visit" when it should state annual dental "examination" . The Healthy People 2020 Survey uses "annual dental examination" as language for both target and target setting method. The term "visit " is unclear as it could include planned or emergency dental treatment. Furthermore, the CDC Diabetes Care Schedule's states dental exam and cleaning at least once per year, and the ADA Standards of Medical Care in Diabetes (2020) state annual dental visit, but in the context of preventive care services (pp 47-48). The author notes this limitation in the discussion section of the study, "There was also no information indicating whether dental visits were for preventive care, emergencies, or planned dental treatment." The author confuses the reader by using the word "visit" and not "examination" throughout the paper, making the discussion unclear and weakens the study's conclusions. Revisiting to the CDC and ADA literature cited in the study and using the correct language throughout the paper may make the submission acceptable for publication.

Response: We thank the reviewer for the good points. We have further clarified throughout the manuscript whether dental visits were for any reason or for preventive examinations.

Reviewer #2:

The methodology and statistical analysis are rigorous. Authors must put the website address on which the BRFSS database is accessible. The authors need to elaborate on the procedure of matching the sample to the US population. The BMI must be defined.

Response: We appreciate the positive feedback from the reviewer. We have added the website information to access the BRFSS database, defined the BMI abbreviation, and added more details to the methods section to elaborate on the procedure of matching the sample to the US population.

---

## [Editor Report · Decision Letter 1]

21 Apr 2021

US adults with diabetes mellitus: Variability in oral healthcare utilization

PONE-D-21-04886R1

Dear Dr. Yu,

We’re pleased to inform you that your manuscript has been judged scientifically suitable for publication and will be formally accepted for publication once it meets all outstanding technical requirements.

Kind regards,

Denis Bourgeois

Academic Editor

PLOS ONE
---

## [Editor Report · Acceptance letter]

27 Apr 2021

PONE-D-21-04886R1 

US adults with diabetes mellitus: Variability in oral healthcare utilization 

Dear Dr. Yu:

I'm pleased to inform you that your manuscript has been deemed suitable for publication in PLOS ONE. Congratulations! Your manuscript is now with our production department. 

Kind regards, 

on behalf of

Professor Denis Bourgeois 

Academic Editor

PLOS ONE